# Steady-State Visual-Evoked-Potential–Driven Quadrotor Control Using a Deep Residual CNN for Short-Time Signal Classification

**DOI:** 10.3390/s25154779

**Published:** 2025-08-03

**Authors:** Jiannan Chen, Chenju Yang, Rao Wei, Changchun Hua, Dianrui Mu, Fuchun Sun

**Affiliations:** 1School of Electrical Engineering, Yanshan University, Qinhuangdao 066004, China; cjn@ysu.edu.cn (J.C.); yangchenju@stumail.ysu.edu.cn (C.Y.); cch@ysu.edu.cn (C.H.); mdr@stumail.ysu.edu.cn (D.M.); 2Department of Computer Science and Technology, Tsinghua University, Beijing 100084, China; fcsun@mail.tsinghua.edu.cn

**Keywords:** brain–computer interface, deep convolutional neural network, quadrotor, short time window, steady-state visual evoked potentials

## Abstract

In this paper, we study the classification problem of short-time-window steady-state visual evoked potentials (SSVEPs) and propose a novel deep convolutional network named EEGResNet based on the idea of residual connection to further improve the classification performance. Since the frequency-domain features extracted from short-time-window signals are difficult to distinguish, the EEGResNet starts from the filter bank (FB)-based feature extraction module in the time domain. The FB designed in this paper is composed of four sixth-order Butterworth filters with different bandpass ranges, and the four bandwidths are 19–50 Hz, 14–38 Hz, 9–26 Hz, and 3–14 Hz, respectively. Then, the extracted four feature tensors with the same shape are directly aggregated together. Furthermore, the aggregated features are further learned by a six-layer convolutional neural network with residual connections. Finally, the network output is generated through an adaptive fully connected layer. To prove the effectiveness and superiority of our designed EEGResNet, necessary experiments and comparisons are conducted over two large public datasets. To further verify the application potential of the trained network, a virtual simulation of brain computer interface (BCI) based quadrotor control is presented through V-REP.

## 1. Introduction

Brain–computer interface (BCI) is a system that measures activity of a human’s or animal’s central nervous system (CNS) and converts it into artificial output that replaces, restores, enhances, supplements, or improves natural CNS output [1,2,3,4,5]. BCI is usually divided into invasive BCI and non-invasive BCI based on whether the detection device invades biological tissue. Given the substantial risks associated with invasive brain–computer interfaces (BCIs) to humans, non-invasive BCIs have garnered extensive research attention. In non-invasive BCI, commonly used signals include motor imagery (MI) [6], event-related potential (ERP) [7], and steady-state visual evoked potential (SSVEP) [8,9].

The SSVEP is obtained after the signal is collected by electroencephalograph (EEG) cap and amplified by an amplifier. Thus, this paper focuses on SSVEP-based BCI research, where participants fixate on a flickering visual target (e.g., point or geometric figure) displayed at a constant frequency on a liquid crystal display (LCD) screen.

In order to determine the true intention of the subject from the SSVEP signal, the signal needs to be decoded. This kind of research is challenging due to the low signal-to-noise ratio of EEG signals and their susceptibility to subject attention. Therefore, many scholars devote themselves to solving this challenging problem, and many methods have been put forward. Canonical correlation analysis (CCA) [10,11] is one of the well-established methods for SSVEP detection, which quantifies the canonical correlation between recorded neural data and predefined stimulus templates. However, its performance significantly degrades when processing short-time window signals. To improve performance, the modified CCA method called CCA-M3 is proposed in [12,13,14]. The methods of CCA and CCA-M3 are similar, which are used to identify signals by establishing the relationship between SSVEP signal and another contrast signal. The only difference is that the contrast signal in the CCA-M3 method is composed of training data. Although CCA-M3 improves classification accuracy for short-time-window signals, it introduces challenges in aligning these time windows.

To significantly boost recognition performance, many scholars focus on deep learning, especially convolutional neural networks (CNNs). The CNN has become a very popular deep learning method for its outstanding performance for image classification problems [15,16]. Recently, the CNN’s ability to learn invariant features has shown promise in EEG signal analysis [17,18,19,20].

However, most of these methods depend on frequency domain features as network input after fast Fourier Transform (FFT) [21,22], because it has obvious features (spikes) in a long time window (FFT-CNN). Due to the limitations of Fourier transform and other similar methods in extracting frequency features from short-time-window signals while preserving time-domain information (e.g., phase characteristics), we propose a time-domain CNN (tCNN) that operates directly on raw temporal signals across varying window lengths. In [23,24], two distinct CNN architectures based on time-domain signals were introduced. Nevertheless, validation trials were carried out on datasets with only a small number of subjects in [23]. For [24], the designed network does not consider the impact of different time windows on the results, which may limit its applicability. In [25,26], a CNN method is also adopted. Before feeding data into the network, filter banks are used to pre-filter the signals, thereby enhancing the network’s ability to efficiently extract task-relevant features, such as the frequency characteristics of the fundamental wave and its harmonics, from short-time-window signals.

Inspired by the preceding analysis and motivated by the findings that the network performance improves as the network depth increases, a novel network called EEGResNet is designed for the classification of steady-state visual evoked potentials, and this network was further applied to the control of quadrotors. The main contributions of this article can be summarized as follows:The network we designed has multiple CNN layers, so it has a strong feature learning ability. At the same time, the designed network makes use of ResNet’s residual connection skills; thus, it can automatically adjust the network structure according to the data characteristics.Our designed network was validated on two large public datasets [27]; thus, the results are highly persuasive. The results show that the EEGResNet specifically surpasses state-of-the-art methods in short-time-window analysis (64 sp, 128 sp, for example (sp: sampling
point)).A BCI-based quadrotor control scheme is proposed, which demonstrates the potential of the designed network in practical applications. Because the shorter time window means the system identification lag time is shorter, it is easier to achieve the real-time control of the equipment.

The remainder of this paper is organized as follows. Section 2 comprehensively describes the proposed approach. To rigorously evaluate the efficacy of our proposed network, Section 3 presents extensive experimental studies. Section 4 presents a quadrotor drone controlled by a BCI, validating the practical implementation potential of the proposed control scheme. Finally, a comprehensive summary is presented in Section 5.

## 2. Proposed Method

This section introduces the EEGResNet framework for SSVEP classification and its integrated BCI-based quadrotor control methodology. Our source code for EEGResNet will be available on Github: https://github.com/raow923/EEGResNet (accessed on 31 July 2025).

### 2.1. Public Dataset

To validate the accuracy and superiority of the proposed network, we trained it on two public datasets, “Session01” and “Session02”, both sampled at 1000 Hz using the international 10–20 system electrode configuration. Each public dataset contains 54 subjects, including 29 males. The subjects range in age from 24 to 35, and 16 of them have experience with BCI testing. The two datasets were collected from the same participants during separate experimental sessions on different dates.

Each public dataset includes a training data subset (“EEG_SSVEP_train”) and test data subset (“EEG_SSVEP_test”), and each subset contains 100 trials. Each trial consists of a 4 s reminder phase (preparation), followed by a 4 s target flashing phase (stimulation), and ends with a 2 s rest phase. The four stimulation targets are positioned at the right, top, left and bottom of the screen, flashing at 8.57 Hz, 5.45 Hz, 12 Hz, and 6.67 Hz, respectively. Thus, each target is repeated 25 times (category 4×25=100 trials). For more detailed information, please refer to reference [27].

### 2.2. Data Preprocessing

The EEG signals in the occipital region have a high signal-to-noise ratio. Nine electrode signals (Pz, P1, P2, PO3, PO4, POz, Oz, O1, and O2) near the occipital region were selected to train and verify the network. To effectively leverage both fundamental and harmonic information, the filter bank design prioritizes two critical factors: (1) the frequency ranges of the fundamental and harmonics and (2) the signal-to-noise ratio (SNR) of the harmonics. Therefore, the filter bank integrates four sub-filters (with passbands of 19–50 Hz, 14–38 Hz, 9–26 Hz, and 3–14 Hz, respectively) to capture the specific harmonic information of all stimulation targets while discarding data at frequencies above 50 Hz due to low SNR [28]. Consequently, during preprocessing, raw signals from the nine channels undergo downsampling to 250 Hz to reduce computational complexity, followed by processing with four sixth-order Butterworth bandpass filters.

### 2.3. Network Structure

Figure 1 illustrates the overall architecture of EEGResNet. To demonstrate the network architecture in greater detail, a time window spanning 64 sp (sampling points) was employed as an illustrative case.

EEGResNet: Firstly, in order to reasonably improve the computing efficiency, the raw data with a sampling frequency of 1000 Hz is down-sampled to 250 Hz. By introducing a time window of length 64 sp to crop the raw data, we obtain the data with the shape (9×P,P=64). In view of the time-domain characteristics of time window signals being easy to distinguish, we designed a set of filters containing four sixth-order Butterworth filters with 19–50 Hz, 14–38 Hz, 9–26 Hz, and 3–14 Hz bandpass ranges to filter the data. And the data obtained by the four filters is directly aggregated together, and the data shape becomes (4×9×P).

Secondly, the features of the obtained data are extracted through Conv1, which includes nn.Conv2d (4, 16, (9,3), (1,1), (0,1)) and nn.BatchNorm2d (16). Then, to efficiently learn features, a deep neural network with a depth of six layers was designed: Layer 1 (inchannel =16, outchannel =16, stride =2), Layer 2 (inchannel =16, outchannel =16, stride =2), Layer 3 (inchannel =16, outchannel =32, stride =2), Layer 4 (inchannel =32, outchannel =32, stride =2), Layer 5 (inchannel =32, outchannel =64, stride =2), and Layer 6 (inchannel =64, outchannel =64, stride =2). Between layers, a dropout layer with 0.25 rate is added to prevent overfitting. After the feature learning of the six-layer neural network, the data shape becomes (64,1,P/64).

Thirdly, to understand each structure of Layers 1–6 in detail, Figure 1b presents a detailed diagram. For each layer (inchannel, outchannel, stride), there are two paths: a **Main Path** represented by a solid line and a **Short-cut Path** represented by a dashed line. The **Main Path** consists of nn.Conv2d (inchannel, outchannel, (1,3), (1, stride), (0,1)), nn.BatchNorm2d (outchannel), nn.Conv2d (outchannel, outchannel, (1,3), (1,1), (0,1)), and nn.BatchNorm2d (outchannel). There are two cases of short-cuts: When the condition (stride != 1 or inchannel != outchannel) holds, the **Short-cut Path** is composed of nn.Conv2d (inchannel, outchannel, (1,1), (1, stride)) and nn.BatchNorm2d (outchannel). If the condition is not true, it is composed of an identity operation. Then, the data generated by the **Main Path** and the data generated by the **Short-cut Path** are then added directly, and activated by the activation function nn.ELU (alpha=1).

In the end, the learned features are flattened into the shape of (1×P), and the network output is generated through a fully connected network nn.Linear(*P*, 4).

**Remark** **1.**
*For the EEGResNet network presented in Figure 1, a short time window of 64 sp is employed as an example. For time windows of other lengths such as 128, 256, and 512, our network can also adapt, and it only needs to adjust P to the corresponding time window length. This flexibility constitutes a key distinctive feature against existing methods.*


## 3. Experiment Results and Discussion

In this section, we provide a detailed introduction to the training process and perform extensive comparative experiments to validate the efficacy of the EEGResNet architecture.

### 3.1. Training Processes

The proposed network was written in Python, and the network structure was built based on Pytorch 1.10. The Python interpreter version was 3.6.15. For each public dataset, there were 100 sets of training trials and 100 sets of test trials, and each trial contained 4 s stimulus data. In the training process, we divided 100 sets of training trials into 9:1 groups, among which 90 sets were used to train the network, and the other 10 sets were employed to validate the network. After the validation process, the trained network was tested with 100 sets of test trials. In order to have enough data to train, validate, and test the network, we used a method called the random initialization sliding window to increase the amount of data appropriately. See Remark 2 for details. Thus, we prepared 2000 data points for training, 2000 data points for validation, and 2000 data points for testing. We trained the network on GeForce GTX TITAN X, and selected the CrossEntropyLoss and Adam (with learning rate 8 × 10^−4^ and weight decay 0.01) as a loss function and optimizer, respectively. The batch size and iteration times were set to 250 and 2000, respectively. These hyperparameters were determined via grid search.

**Remark** **2.**
*Given a time window with the range [0.14 + r, 0.14 + r + d] s [28], if r is a random positive number and d is a fixed positive number, then we call this method the random initialization sliding window. In this paper, r is a random number in the range [0, 4 − d], where 4 s represents the time length of the original data in each trial, and d s is the time length of the time window. Training samples were subsequently extracted from the original dataset through the randomly initialized sliding window.*


### 3.2. Baseline Networks

To validate the accuracy and superiority of the proposed network, this study conducted comparative experiments against state-of-the-art baselines. Seven baselines, including CNN, CCA, CCA-M3, tCNN, FFT-CNN, Compact-CNN, and Resnet-18 were selected, and the necessary descriptions are given as follows.

CCA: [10] The SSVEP signal generated by the stimulus has a specific relationship with the target signal in the frequency domain. CCA identifies SSVEP signals by establishing this relationship.CCA-M3 [12]: The methods of CCA and CCA-M3 are similar, which are used to identify signals by establishing the relationship between SSVEP signal and another contrast signal. The only difference is that the contrast signal in the CCA-M3 method is composed of training data.FFT-CNN [22]: The FFT-CNN method firstly uses Fast Fourier Transform to extract frequency features, and then these frequency features are input into the CNN for feature learning to achieve signal identification.Compact-CNN [24]: The Compact-CNN is composed of CNN network layers and pooling layers. Unlike FFT-CNN, which requires frequency transformation, Compact-CNN uniquely employs raw time-domain signals as its input, eliminating preprocessing overhead.tCNN [28]: Similarly to the Compact-CNN, tCNN directly uses signals in the time domain as the input of the network. To simplify the network, tCNN removes the pooling layer in Compact-CNN.Resnet-18: It employs four stages of residual blocks, where short-cut connections directly add the input to the output within each block.CNN [23]: It serves as an end-to-end network architecture capable of directly classifying SSVEP stimuli from dry EEG waveforms without manual feature extraction.

### 3.3. Results

In this subsection, two sets of experiments were carried out, and the corresponding results are presented. The first set of experiments verified the effectiveness and superiority of our designed network with the shortest time window. The second set of experiments explored the influence of time window length variation on network performance.

**The first set of experiments:** The length of the time window was set to 64 sp. Table 1 presents the mean accuracy and F1-score of 54 participants on the public datasets “Session01” and “Session02” during the test phase, respectively. As shown in the table, it can be clearly observed that our designed EEGResNet network has obvious advantages. The model demonstrates absolute gains of at least 6.0% in accuracy and 5.6% in F1-score over baseline models. Additionally, we conducted a detailed analysis of the sensitivity, specificity, and F1-score for all models across the four classes, as presented in Table 2 and Table 3. As shown in the tables, EEGResNet achieves consistently high performance across all three metrics, surpassing most baseline models. Across both datasets, EEGResNet demonstrates the highest sensitivity to Class 1 and Class 2, achieving 83.54% and 84.16% on “Session01” and 86.32% and 85.75% on “Session02”. For “Session01”, Class 1 and Class 2 achieve specificity values of 95.92% and 96.20%, with F1-scores of 85.29% and 86.11%, respectively. For “Session02”, Class 1 and Class 2 achieve specificity values of 95.84% and 97.08%, with F1-scores of 86.86% and 88.22%, respectively.

**The second set of experiments:** The lengths of the time window were set to 64 sp, 128 sp, and 256 sp, respectively. Figure 2 depicts the mean accuracy and standard deviation (across 54 subjects) of all models across varying time window lengths for both the “Session01” and “Session02” datasets. From the experimental results depicted in the figures, it is evident that the average accuracy of all networks tends to improve as the time window length increases; however, the rate of this improvement progressively diminishes. Additionally, the EEGResNet network consistently surpasses other baseline models in accuracy across all time window lengths, demonstrating notable advantages. Our average accuracy is at least 6, 1, and 0.5 percentage points higher than the average accuracy of other networks at 64 sp, 128 sp, and 256 sp time windows, respectively. To further comprehensively evaluate our model, we conducted paired t-tests comparing EEGResNet with each baseline model across two datasets, as illustrated in Figure 3. As time window length increases, *p*-values rise but remain below 0.05, confirming the significant difference between our model and baseline models. Integrated analysis of Table 1 and Figure 2 and Figure 3 reveals that the model achieves higher accuracy than baseline models across both datasets, with statistically significant differences (*p* < 0.05). These results showcase the superior robustness and generalization capabilities of EEGResNet.

Collectively, the proposed EEGResNet demonstrates superior performance over baseline networks in short-time-window signal recognition. This capability contributes to significantly reduced lag times during signal processing, highlighting its stronger potential for real-time applications.

## 4. Application

In this section, a quadrotor control scheme based on BCI is presented, and the application scenario is implemented on V-REP 4.2.0 or CoppeliaSim 4.2.0 software.

### 4.1. Scenario Design

As shown in Figure 4, the designed scenario consists of four non-movable objects with different colors and a movable quadrotor: Red Cylinder, Blue Cuboid, Black Sphere, White Cuboid, and the quadrotor. The position parameters of non-movable objects Red Cylinder, Blue Cuboid, Black Sphere, and White Cuboid were [1, 0, 0.12] m, [0, −1, 0.12] m, [−1, 0, 0.12] m, and [0, 1, 0.12] m, respectively. The initial position parameter of the movable quadrotor was [0, 0, 3] m. The quadrotor had a fixed flight altitude during the whole operation period. The control target of the designed scenario was to control the quadrotor to fly directly over a specific object according to the classification results of the trained network EEGResNet.

### 4.2. BCI-Based Control

The overall control strategy for the designed scenario is shown in Figure 5, and is composed of test signals, the trained network EEGResNet, the quadrotor control interface, and the designed scenario.

We utilized the Remote API as the communication protocol between V-REP and the Python control interface. In V-REP, the Remote API server-side script is embedded within the simulation scene to listen for client connection requests and process incoming commands. Within the Python control interface script, the EEGResNet network output is algorithmically analyzed to determine the target position. For each target position, the Python script transmits control commands containing its coordinates to the quadrotor via the Remote API.

In this experiment, there were four test signals [$x_{0}$, 0], [$x_{1}$, 1], [$x_{2}$, 2], and [$x_{3}$, 3], which were cropped from the public dataset “Session01” according to a short time window with a length of 64 sp. In order to efficiently identify the test signals, the EEGResNet architecture, pre-trained on the “Session01” dataset, was leveraged to decode the signals. The four test signals indicate four different moving targets, which means if EEGResNet outputs equal to [1, 0, 0, 0], [0, 1, 0, 0], [0, 0, 1, 0], and [0, 0, 0, 1], then the quadrotor will fly to the target position 0, 1, 2, and 3, respectively. The flight dynamic diagram of this application is shown in Figure 6, and as can be seen from the figure, our designed network can decode the test signal accurately and control the quadrotor to fly directly above the desired target.

**Remark** **3.**
*In order to enable the quadrotor to recognize the network output information and make corresponding movements, a quadrotor control interface must be designed. This interface is written in Python and connects to the designed scenario through the official communication interface provided by V-REP. In simple terms, the quadrotor control interface includes one conditional statement and four instructions. Throughout the experiment, the quadrotor maintains a consistent flight altitude while proceeding toward the predefined target. We will make the Python code for the gesture control interface open source on GitHub (https://github.com/raow923/EEGResNet (accessed on 31 July 2025), with further details to follow.*


## 5. Conclusions

In this paper, we propose an enhanced EEGResNet architecture derived from existing frameworks to boost short-time-window SSVEP recognition performance, concurrently developing an application scenario to explore its potential applications. First, a set of filters is used to extract time-domain features. Then, the obtained features are further learned by a six-layer convolutional neural network with residual connections. Finally, the network output is generated through an adaptive fully connected layer. The experimental results on two large public datasets demonstrate that the proposed EEGResNet outperforms state-of-the-art methods, especially at short time windows. In practical BCI applications, reduced time windows enable faster signal identification rates, which are critical for real-time performance. Thus, we devised an application scenario wherein our trained network is utilized to facilitate direct SSVEP-based control of a quadrotor, thereby showcasing the practical potential of our proposed network. However, the scenario we designed needs to know the position information of objects in advance, which is difficult to achieve in practice. Therefore, we will use object detection algorithms combined with GPS and depth cameras to solve this problem in the future.

## Figures and Tables

**Figure 1 sensors-25-04779-f001:**
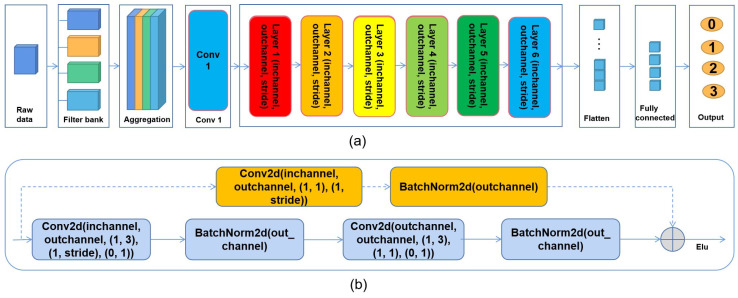
(**a**) The specific architecture of the EEGResNet network; (**b**) the specific structure of the network at Layer *x*, *x*
∈{1,2,3,4,5,6}. In this figure, Conv stands for convolution.

**Figure 2 sensors-25-04779-f002:**
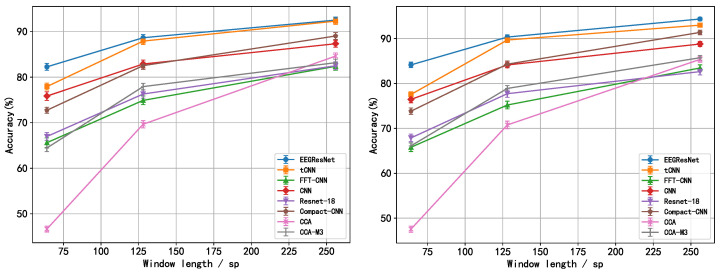
The average accuracy and standard deviation of EEGResNet and baseline networks on the public datasets “Session01” and “Session02” are shown, with the vertical axis representing average accuracy (mean ± std %) and the horizontal axis indicating the time window. (**Left**) “Session01”; (**right**) “Session02”.

**Figure 3 sensors-25-04779-f003:**
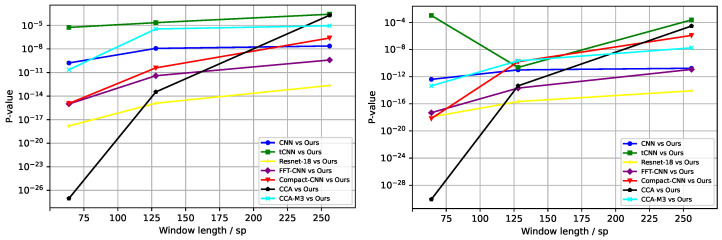
The *p*-values for EEGResNet and baseline networks on the public datasets “Session01” and “Session02” are shown, with the vertical axis indicating the *p*-value and the horizontal axis representing the time window. (**Left**) “Session01”; (**right**) “Session02”.

**Figure 4 sensors-25-04779-f004:**
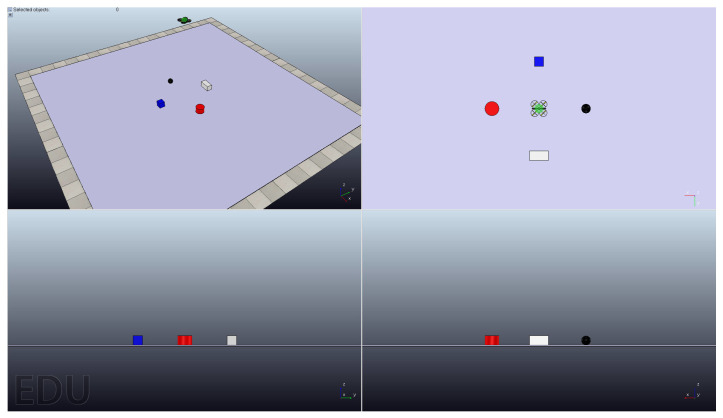
The designed scenario, composed of four non-movable objects with different colors and a movable quadrotor: Red Cylinder, Blue Cuboid, Black Sphere, White Cuboid, and the quadrotor.

**Figure 5 sensors-25-04779-f005:**
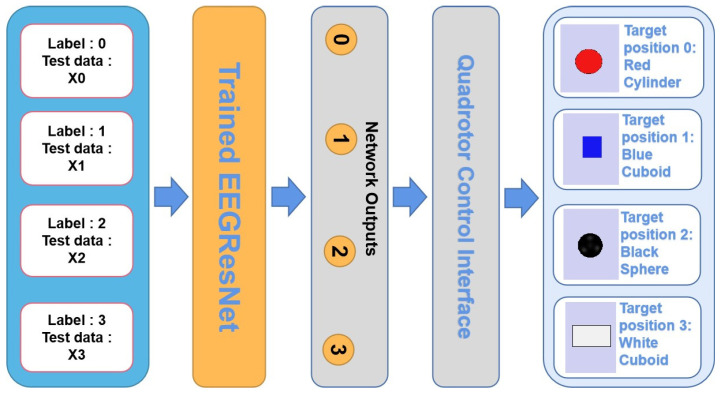
The integrated control strategy for quadrotor systems based on BCI consists of four test signals, the trained EEGResNet, the quadrotor control interface, and the designed scenario. The network outputs 0, 1, 2, and 3, which correspond to the one-hot vectors [1, 0, 0, 0], [0, 1, 0, 0], [0, 0, 1, 0], and [0, 0, 0, 1] respectively.

**Figure 6 sensors-25-04779-f006:**
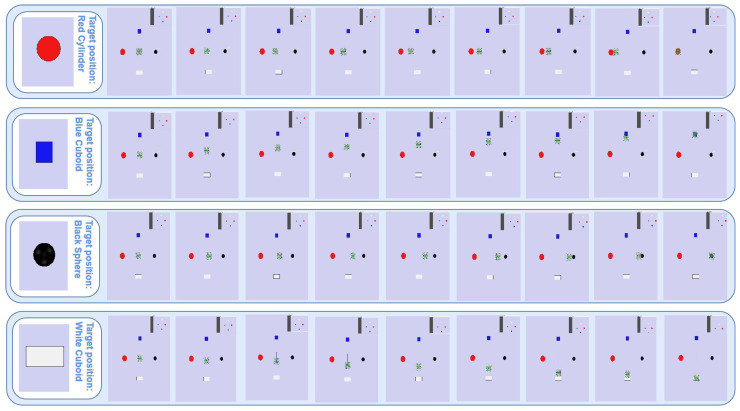
The application outcomes of the quadrotor control scheme based on BCI. The sequence diagram visualizes the trajectory tracking process from the initial position to the target position.

**Table 1 sensors-25-04779-t001:** The average accuracy (mean/std %) and F1-score (mean/std %) of all models on two public datasets with a 64 sp time window.

Method	Session01		Session02	
	Mean/std	F1/std	Mean/std	F1/std
FFT-CNN	65.62/2.05	67.99/1.89	65.80/1.92	68.06/1.78
Compact-CNN	72.74/1.70	73.55/1.67	73.84/1.43	74.62/1.43
tCNN	76.34/1.64	77.93/1.51	77.56/1.38	79.10/1.26
Resnet-18	66.98/1.94	69.23/1.81	67.89/1.79	69.26/1.68
CNN	75.85/1.78	77.60/1.66	76.47/1.54	78.27/1.41
CCA	46.66/1.28	42.55/–	47.55/1.36	32.75/–
CCA-M3	64.44/1.60	74.27/–	66.11/1.41	68.15/–
EEGResNet (Ours)	82.27/1.56	83.55/1.44	84.15/1.24	85.39/1.14

**Table 2 sensors-25-04779-t002:** The sensitivity (%), specificity (%), and F1-score (%) of all models for the four classes on the “Session01” dataset with a 64 sp time window.

Classes	Metrics	FFT-CNN	Compact-CNN	tCNN	Resnet-18	CNN	CCA	CCA-M3	Ours
	Sensitivity	71.97	86.19	79.73	71.35	78.19	37.50	76.92	83.54
Class 1	Specificity	90.48	90.60	94.75	90.93	93.25	97.37	90.24	95.92
	F1-score	71.77	79.94	81.50	71.99	78.88	52.17	74.07	85.29
	Sensitivity	68.47	72.35	78.71	71.78	77.80	25.00	62.50	84.16
Class 2	Specificity	90.10	95.01	95.37	89.42	94.80	84.21	94.74	96.20
	F1-score	69.09	77.25	81.75	70.66	80.41	30.77	71.43	86.11
	Sensitivity	62.32	77.11	72.98	63.71	74.02	28.57	78.57	78.75
Class 3	Specificity	87.98	83.74	91.81	88.29	92.00	91.49	87.50	94.36
	F1-score	62.82	68.67	73.91	63.86	74.75	30.77	73.33	80.50
	Sensitivity	59.95	51.65	74.98	61.30	73.88	86.67	81.82	81.57
Class 4	Specificity	85.68	92.85	86.86	87.47	87.93	53.85	93.02	89.52
	F1-score	59.10	59.78	69.98	61.58	70.30	56.52	78.26	76.61

**Table 3 sensors-25-04779-t003:** The sensitivity (%), specificity (%), and F1-score (%) of all models for the four classes on the “Session02” dataset with a 64 sp time window.

Classes	Metrics	FFT-CNN	Compact-CNN	tCNN	Resnet-18	CNN	CCA	CCA-M3	Ours
	Sensitivity	71.63	86.29	82.96	74.20	80.19	00.00	89.47	86.32
Class 1	Specificity	90.79	96.46	94.31	91.15	93.19	93.18	82.86	95.84
	F1-score	71.97	87.74	82.94	74.06	80.00	00.00	80.95	86.86
	Sensitivity	69.70	83.57	81.16	71.48	78.86	44.44	60.00	85.75
Class 2	Specificity	90.41	93.90	95.38	91.06	95.36	1.00	95.45	97.08
	F1-score	70.23	82.73	83.31	72.12	81.83	61.54	66.67	88.22
	Sensitivity	60.73	80.03	72.85	63.64	74.21	33.33	66.67	81.75
Class 3	Specificity	88.35	90.15	92.43	88.43	92.45	76.92	90.48	95.08
	F1-score	62.00	76.54	74.43	64.02	75.36	34.48	66.67	83.07
	Sensitivity	61.68	67.41	75.38	62.89	75.03	63.64	53.85	83.65
Class 4	Specificity	85.05	92.04	88.71	86.81	88.45	48.84	90.24	91.08
	F1-score	59.74	70.25	71.98	62.09	71.50	35.00	58.33	79.69

## Data Availability

Restrictions apply to the availability of these data. Data were obtained from Lee, M.-H. et al. (GigaScience, 2019) and are available from the corresponding author of that publication with their permission.

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
