# Peer review of "Steady-State Visual-Evoked-Potential–Driven Quadrotor Control Using a Deep Residual CNN for Short-Time Signal Classification"

_sensors, 2025, doi:10.3390/s25154779_

Round 1
Reviewer 1 Report
Comments and Suggestions for Authors
This study presents a deep convolutional neural network, EEGResNet, based on the concept of residual connections to improve the classification of steady-state visual evoked potentials (SSVEP). However, the paper lacks sufficient methodological clarity and has several critical issues that must be addressed:
- The protocols for Database 1 and Database 2 are insufficiently described. The Methods section should provide:
- The number of classes and their corresponding labels
- The number of trials per class for both the training and testing phases
- Any differences in the experimental protocols or EEG acquisition systems between the two datasets
Additionally, it is unclear what the subjects are doing during the 4-second target flashing phase. This should be explicitly stated. - The rationale for choosing four sixth-order Butterworth bandpass filters with passbands of 19–50 Hz, 14–38 Hz, 9–26 Hz, and 3–14 Hz is not clearly justified. These ranges do not correspond to conventional EEG frequency bands (delta, theta, alpha, beta, gamma). Please provide a justification for these selections, particularly in the context of SSVEP.
- In Table 1 titled "Mean Accuracy (Mean ± SEM%) Across All Models on Public Datasets with a 64 Time Window", the statistical meaning of the reported average and standard deviation is unclear. Given that 90% of the data is used for training and 10% for validation, and a separate folder is used for testing, it must be clarified:
- How many repetitions or cross-validations were done to compute the mean and standard deviation?
- What does "Mean ± std" represent in Figures 2 and 3?
Moreover, to better assess model performance, sensitivity, specificity, and F1-score for each class and the performance for training on one database (1 or 2) and testing on the other should be included.
- The description of the four classes used in Figure 1 for the quadrotor control scheme is vague. The paper should explain:
- The control protocol and how each class corresponds to a specific command
- How subjects interact with the system during tasks
- The number of subjects who participated in the quadrotor control experiment
- Whether the pretrained models from Databases 1 and 2 were used in this section
- The performance metrics achieved during quadrotor control
Due to the above significant omissions and lack of methodological clarity throughout the manuscript, I recommend a reject and resubmit decision. The authors are encouraged to revise and resubmit the manuscript with detailed methodological explanations, justified design choices, and comprehensive performance metrics.
Author Response
|
Comments 1: The protocols for Database 1 and Database 2 are insufficiently described. The Methods section should provide: The number of classes and their corresponding labels The number of trials per class for both the training and testing phases Any differences in the experimental protocols or EEG acquisition systems between the two datasets Response 1: Thank you for pointing this out. We have provided detailed descriptions for Dataset 1 and Dataset 2. (1) Each dataset is divided into a training subset ("EEG_SSVEP_train") and a testing subset ("EEG_SSVEP_test"). The dataset comprises four stimulation targets (i.e., four classes (the flicker frequencies are 8.57 Hz, 5.45 Hz, 12 Hz, and 6.67 Hz) with labels 1, 2, 3, and 4), with each class having 25 trials. (2) Both the training and testing subsets contain 100 trials. Thus, our network employs 25 trials per class during both the training and testing stages. (3) The two datasets (“session01” and “session02”) are experimental data collected from the same group of participants on different dates, both sampled at a frequency of 1000 Hz using the international 10-20 system electrode configuration. (4) During the 4-second target flickering phase, the subjects fixate on the flickering target. There is a detailed explanation in the section "2.1 Public Dataset" on page 3. |
|
Comments 2: The rationale for choosing four sixth-order Butterworth bandpass filters with passbands of 19–50 Hz, 14–38 Hz, 9–26 Hz, and 3–14 Hz is not clearly justified. These ranges do not correspond to conventional EEG frequency bands (delta, theta, alpha, beta, gamma). Please provide a justification for these selections, particularly in the context of SSVEP. |
|
Response 2: We will provide detailed explanations for the four filters with different bandpass ranges. To effectively leverage both fundamental and harmonic information within the dataset, the design of the filter bank mainly considers two factors: 1) the range of the fundamental and harmonic frequencies, and 2) the signal-to-noise ratio (SNR) of the harmonics. To capture the specific harmonic information of the four stimulation targets (the flicker frequencies are 8.57 Hz, 5.45 Hz, 12 Hz, and 6.67 Hz) while discarding low SNR data above 50 Hz, the passbands of the four sub-filters are set to 19-50 Hz, 14-38 Hz, 9-26 Hz, and 3-14 Hz. We have provided the rationale for selecting these four bandpass ranges in the "2.2 Data Preprocessing" section on page 3. The design of the filters is based on the 28th reference. Comments 3: In Table 1 titled "Mean Accuracy (Mean ± SEM%) Across All Models on Public Datasets with a 64 Time Window", the statistical meaning of the reported average and standard deviation is unclear. Given that 90% of the data is used for training and 10% for validation, and a separate folder is used for testing, it must be clarified: How many repetitions or cross-validations were done to compute the mean and standard deviation? What does "Mean ± std" represent in Figures 2 and 3? Response 3: For the mean accuracy and standard deviation in Table 1: Taking the time window of 64 as an example, all models are trained on the two datasets. The accuracy for one subject is determined by taking the maximum accuracy across all epochs. The mean and standard deviation of accuracies for the 54 subjects are then calculated. The results are presented in Table 1. Figure 2 (modified) compares the mean accuracy and standard deviation of the 54 subjects for all models at different time window lengths on the two datasets, respectively. A detailed explanation has been added on page 6, line 220. Comments 4: Moreover, to better assess model performance, sensitivity, specificity, and F1-score for each class and the performance for training on one database (1 or 2) and testing on the other should be included. Response 4: We have accepted your suggestion and added the F1-score and standard deviation of each model on the two datasets in Table 1. Two additional baselines are introduced: ResNet-18 and CNN whose depth matches that of our model. The updated Table 1 is presented on page 6. Our network is trained in an intrasubject setting. Training on one dataset and testing on another represents a cross-subject scenario. Despite sharing the same participants, the collection of the two datasets on different dates may introduce variability in EEG signals due to fluctuations in physiological states (e.g., fatigue, attention). The model trained on Dataset 1 may not effectively adapt to the distribution of Dataset 2. Thus, on the same dataset, we split the training subset (EEG_SSVEP_train) into training and validation sets at a 9:1 ratio and use the entire testing subset (EEG_SSVEP_test) as the test set. This approach is also used in the 28th reference. |
Comments 5: The description of the four classes used in Figure 1 for the quadrotor control scheme is vague. The paper should explain:
The control protocol and how each class corresponds to a specific command
How subjects interact with the system during tasks
The number of subjects who participated in the quadrotor control experiment
Whether the pretrained models from Databases 1 and 2 were used in this section
The performance metrics achieved during quadrotor control
Response 5: We provide the following explanations for the application section:
Fig 5 shows the network's output being compiled into the quadrotor's target position via its control interface. We extract four-category data from the public dataset "session01", employing a network trained exclusively on this dataset for signal classification. No online experiments are conducted (for detailed explanations, see line 270 on page 8).
During the experiment, we use a remote API as the communication protocol between the V-REP and the Python control interface. In V-REP, the remote API server-side script is loaded into the simulation scene. It listens for connection requests from the client and processes the commands sent by the client. In the Python control interface script, the target position is determined by decoding the output from the EEGResNet network. For each target position, the Python script uses the remote API to send control commands to the quadrotor, including the coordinates of the target position. Detailed explanations regarding the quadrotor flight control and the relevant communication protocols have been incorporated on line 263, page 8.
Our network achieves accurate four-category signal classification. Control commands containing the coordinates of the target position are sent to the quadrotor based on the network's output, enabling it to accurately fly to the target position, as shown in Figure 6 on page 9.

Reviewer 2 Report
Comments and Suggestions for Authors
This article introduces EEGResNet, a deep residual convolutional neural network designed for the classification of short‐time‐window SSVEPs, which begins with a four-band Butterworth filter bank to extract time-domain features and proceeds through a six-layer residual CNN before yielding class outputs via an adaptive fully connected layer. The network is trained and evaluated on two large public SSVEP datasets, each comprising 54 subjects. Results demonstrate that EEGResNet achieves higher mean accuracy—particularly with 64 sampling-point windows—than five baseline methods including CCA, CCA-M3, FFT-CNN, Compact-CNN, and tCNN. Furthermore, the paper describes a V-REP-based simulation in which EEGResNet’s outputs are used to control a quadrotor’s flight to one of four colored targets, illustrating the model’s potential for real-time brain–computer interface applications. This article has its significance; however, I have the following major potential concerns to be addressed:
- The claimed novelty of leveraging residual connections in EEGResNet to “automatically adjust the network structure” overlaps substantially with standard ResNet architectures; consider providing an ablation study contrasting EEGResNet against a plain deep CNN of equal depth as well as against a ResNet-18 baseline to isolate the effect of the residual pathways.
- The use of a random initialization sliding-window to expand data (generating 2000 samples per split) risks temporal overlap between training, validation, and test sets; re-design the data splits with nonoverlapping windows or employ a leave-one-trial-out cross-validation to prevent potential data leakage.
- Reporting only mean accuracy and SEM across subjects (e.g., 82.16 ± 2.05%) without formal significance testing limits confidence in performance gains; perform paired statistical tests (e.g., Wilcoxon signed-rank or paired t-tests) between EEGResNet and each baseline and report effect sizes and p-values.
- Hyperparameters such as learning rate = 8e−4, weight decay = 0.01, dropout = 0.25, batch size = 250, and 2000 iterations are stated without justification; include a systematic hyperparameter search (e.g., grid or Bayesian optimization) and present sensitivity analyses to demonstrate robustness to these settings.
- The four sixth-order Butterworth filter bands (3–14 Hz, 9–26 Hz, 14–38 Hz, 19–50 Hz) are selected without comparison to alternative filter banks; evaluate different filter-bank configurations or learnable filter layers to substantiate the chosen frequency ranges.
Author Response
|
Comments 1: The claimed novelty of leveraging residual connections in EEGResNet to “automatically adjust the network structure” overlaps substantially with standard ResNet architectures; consider providing an ablation study contrasting EEGResNet against a plain deep CNN of equal depth as well as against a ResNet-18 baseline to isolate the effect of the residual pathways. Response 1: Following this suggestion, we add ResNet-18 and a depth-matched CNN as baseline networks in experiments, alongside F1-score for comprehensive evaluation. Results are summarized in Table 1. The table shows that our network outperforms other models on both datasets. For detailed data, see Table 1 on page 6. |
|
Comments 2: The use of a random initialization sliding-window to expand data (generating 2000 samples per split) risks temporal overlap between training, validation, and test sets; re-design the data splits with nonoverlapping windows or employ a leave-one-trial-out cross-validation to prevent potential data leakage. |
|
Response 2: The network is trained on the publicly available datasets "Session01" and "Session02". Each dataset is split into a training subset ("EEG_SSVEP_train") and a testing subset ("EEG_SSVEP_test"), each containing 100 trials (4 categories (the flicker frequencies are 8.57 Hz, 5.45 Hz, 12 Hz, and 6.67 Hz), with 25 trials per category). The training subset ("EEG_SSVEP_train") is split into training and validation sets at a 9:1 ratio, while the entire testing subset ("EEG_SSVEP_test") is used as the test set. Therefore, the application of randomly initialized sliding windows for data augmentation ensures no temporal overlap between the training, validation, and test sets. Detailed descriptions are available in "2.1 Public Dataset" on page 3 and "3.1 Training Process" on page 5. The 28th reference adopts a similar approach. Comments 3: Reporting only mean accuracy and SEM across subjects (e.g., 82.16 ± 2.05%) without formal significance testing limits confidence in performance gains; perform paired statistical tests (e.g., Wilcoxon signed-rank or paired t-tests) between EEGResNet and each baseline and report effect sizes and p-values. Response 3: We have implemented this suggestion by incorporating the F1-score metric into Table 1. Additionally, p-value significance curves are plotted across all models under varying time window lengths. Table 1 is on page 6, and the p-value graph is shown as Figure 3 on page 7. Comments 4: Hyperparameters such as learning rate = 8e−4, weight decay = 0.01, dropout = 0.25, batch size = 250, and 2000 iterations are stated without justification; include a systematic hyperparameter search (e.g., grid or Bayesian optimization) and present sensitivity analyses to demonstrate robustness to these settings. Response 4: We obtain the optimal parameter combination of the model through the grid search strategy. We add the reasons for selecting these parameters on line 181 of page 5. Analyzing Figure 2 (accuracy across three time windows) and Figure 3 (p-values comparing our model to others), our model consistently outperforms baseline models with p-values below 0.05. This demonstrates our model's superiority, robustness, and generalization. |
Comments 5: The four sixth-order Butterworth filter bands (3–14 Hz, 9–26 Hz, 14–38 Hz, 19–50 Hz) are selected without comparison to alternative filter banks; evaluate different filter-bank configurations or learnable filter layers to substantiate the chosen frequency ranges.
Response 5: We will provide detailed explanations for the four filters with different bandpass ranges.
To effectively leverage both fundamental and harmonic information within the dataset, the design of the filter bank mainly considers two factors: 1) the range of the fundamental and harmonic frequencies, and 2) the signal-to-noise ratio (SNR) of the harmonics. To capture the specific harmonic information of the four stimulation targets (the flicker frequencies are 8.57 Hz, 5.45 Hz, 12 Hz, and 6.67 Hz) while discarding low SNR data above 50 Hz, the passbands of the four sub-filters are set to 19-50 Hz, 14-38 Hz, 9-26 Hz, and 3-14 Hz.
We have provided the rationale for selecting these four bandpass ranges in the "2.2 Data Preprocessing" section on page 3. The design of the filters is based on the 28th reference.

Reviewer 3 Report
Comments and Suggestions for Authors
The manuscript presents a novel deep convolutional neural network (EERGesNet), designed for the classification of a short-time window SSVEP signals. The architecture begins by using a filter bank of four Butterworth filters to precess a raw time-domain EEG data. The aggregated features are then fed into a six-layer deep residual CNN for classification. The models’ effectiveness is evaluated on two large public datasets, where it is compared against several baseline methods. The authors demonstrate a practical application of developed solution by using trained network to control a quadrotor in a virtual simulation environment (V-REP). The primary contribution of this work is a network that achieves high accuracy on short time windows. Which is crucial for real-time BCI applications.
However, paper could be improved taking into account several suggestions.
- The central claim of the paper is that EEGResNet outperforms other methods. However, to substantiate this, it should be performed statistical tests (such as ANOVA, t-test, etc.) to compare accuracies of all models across all subjects. Please, add such results if possible.
- Please, provide a more detailed explanation of your “random initialization sliding window” algorithm. Please, explicitly state how you ensured that no data from a test subject’s trial could have appeared in any form within the training or validation sets. This is critical to rule out data leakage.
- Please, add a few sentences to the methods section to justify your choice of hyperparameters (or state how they were tuned) and explain the rationale behind selecting the specific frequency bands for your filter bank. This will improve the scientific rigor and reproducibility of your work.
- Please, review the manuscript for minor grammatical errors. Please, moderate the claim of a “novel network” to more accurately reflect that it is a novel adaptation and application of known architecture. Also, fix typo in line 247 – it supposed to be word “equal”, but instead we see ‘eaual’ in the text.
- The quadrotor application, while illustrative, is highly limited. It relies on pre-defined target locations and operates at a fixed altitude. Please, acknowledge these limitations in the Application section, not only in Conclusion, to provide better context.
In general, manuscript contains certain values, and could be accepted for publication after addressing reviewer’s comments.
Comments on the Quality of English Language
Please, double-check English grammar and fix all typos throughout the text.
Author Response
|
Comments 1: The central claim of the paper is that EEGResNet outperforms other methods. However, to substantiate this, it should be performed statistical tests (such as ANOVA, t-test, etc.) to compare accuracies of all models across all subjects. Please, add such results if possible. Response 1: We agree with this viewpoint. Accordingly, p-value plots comparing all models across varying time window lengths are now included in experiments, further highlighting the superiority of our network. The p-value graph is shown as Figure 3 on page 7. Detailed explanations are provided on line 236 of page 7. |
|
Comments 2: Please, provide a more detailed explanation of your “random initialization sliding window” algorithm. Please, explicitly state how you ensured that no data from a test subject’s trial could have appeared in any form within the training or validation sets. This is critical to rule out data leakage. |
|
Response 2: We employ randomly initialized sliding windows to moderately augment the data. A fixed-length time window is randomly selected as a training sample within the range of [0.14+r,0.14+r+d]s. The time window starts at 0.14+r s after the stimulus onset, where r is a random number in the range[0,4-d]. Here, 4 s is the duration of the stimulus, and d is the length of the time window. Training samples are subsequently extracted from the raw data utilizing randomly initialized time windows. Detailed explanations are provided in "Remark 2" on page 5. The 28th reference also follows the same approach. The network is trained on the publicly available datasets "Session01" and "Session02". Each dataset is split into a training subset ("EEG_SSVEP_train") and a testing subset ("EEG_SSVEP_test"), each containing 100 trials (4 categories, with 25 trials per category). The training subset ("EEG_SSVEP_train") is split into training and validation sets at a 9:1 ratio, while the entire testing subset ("EEG_SSVEP_test") is used as the test set. Therefore, the application of randomly initialized sliding windows for data augmentation ensures no temporal overlap between the training, validation, and test sets. Detailed descriptions are available in "2.1 Public Dataset" on page 3 and "3.1 Training Process" on page 5. The 28th reference adopts a similar approach. Comments 3: Please, add a few sentences to the methods section to justify your choice of hyperparameters (or state how they were tuned) and explain the rationale behind selecting the specific frequency bands for your filter bank. This will improve the scientific rigor and reproducibility of your work. Response 3: The hyperparameters in the model are determined using a grid search strategy. We will provide detailed explanations for the four filters with different bandpass ranges [19-50 HZ, 14-38 HZ, 9-26 HZ, 3-14 HZ ]. To effectively leverage both fundamental and harmonic information within the dataset, the design of the filter bank mainly considers two factors: 1) the range of the fundamental and harmonic frequencies, and 2) the signal-to-noise ratio (SNR) of the harmonics. To capture the specific harmonic information of the four stimulation targets while discarding low SNR data above 50 Hz, the passbands of the four sub-filters are set to 19-50 Hz, 14-38 Hz, 9-26 Hz, and 3-14 Hz. We have provided the rationale for selecting these four bandpass ranges in the "2.2 Data Preprocessing" section on page 3. The design of the filters is based on the 28th reference. Comments 4: Please, review the manuscript for minor grammatical errors. Please, moderate the claim of a “novel network” to more accurately reflect that it is a novel adaptation and application of known architecture. Also, fix typo in line 247 – it supposed to be word “equal”, but instead we see ‘eaual’ in the text. Response 4: We concur with this opinion and have revised the grammatical and spelling errors in the manuscript. We have revised the description of the "novel network": we propose an enhanced EEGResNet architecture derived from existing frameworks to boost short time window SSVEP recognition performance, concurrently developing an application scenario to explore its potential applications. The revision is located on line 288 of page 9. |
Comments 5: The quadrotor application, while illustrative, is highly limited. It relies on pre-defined target locations and operates at a fixed altitude. Please, acknowledge these limitations in the Application section, not only in Conclusion, to provide better context.
Response 5: We concur with this viewpoint and have further elaborated in the application section: the quadrotor maintains a consistent flight altitude while proceeding toward the predefined target.
The revision is located in on line 283 of page 8.

Round 2
Reviewer 1 Report
Comments and Suggestions for Authors
The authors addressed most of my comments; however, the following two comments require further elaboration:
1) While I understand the general motivation behind the filter bank design (considering fundamental and harmonic frequencies and SNR), the rationale for the specific passbands (19–50 Hz, 14–38 Hz, 9–26 Hz, and 3–14 Hz) , however, the significant overlap between the sub-filter passbands (e.g., 19–50 Hz and 14–38 Hz) raises the question of how effectively these filters isolate distinct harmonic components of the flicker frequencies. Could you elaborate on why this overlap was deemed necessary and how it impacts the separation of harmonics?
2) Sensitivity, specificity, and F1 score for each of the four classes should be added in Table 1.
Author Response
Comments 1: While I understand the general motivation behind the filter bank design (considering fundamental and harmonic frequencies and SNR), the rationale for the specific passbands (19–50 Hz, 14–38 Hz, 9–26 Hz, and 3–14 Hz) , however, the significant overlap between the sub-filter passbands (e.g., 19–50 Hz and 14–38 Hz) raises the question of how effectively these filters isolate distinct harmonic components of the flicker frequencies. Could you elaborate on why this overlap was deemed necessary and how it impacts the separation of harmonics?
Response 1: We appreciate your thorough attention to our filter design specifications. Indeed, an overlap exists between the passbands, and a detailed explanation will be provided. The target stimulus spans 5.45–12 Hz, with harmonics at 10.9–24 Hz, 16.35–36 Hz, and 21.8–48 Hz. The four designed filters (3–14 Hz, 9–26 Hz, 14–38 Hz, and 19–50 Hz) are engineered to prioritize coverage and enhancement of these specific harmonic components.
1)Broadening the passband ensures that harmonic components which slightly deviate from theoretical frequencies or have weak energy in the actual response can be captured.
2) Individual differences and state fluctuations: SSVEP responses (particularly higher harmonics) and signal-to-noise ratios may exhibit slight fluctuations across different subjects, or even within the same subject under varying timepoints or states. A moderately wider bandwidth ensures robustness by guaranteeing continuous effective coverage of the target frequency.
3)Transition band effect in practical filters: At the passband edges, signal attenuation is gradual. To ensure the target frequency range (especially critical edge frequencies such as 12 Hz or 48 Hz) is positioned in the flat response region of the passband rather than the severely attenuated transition band, the theoretical target range must be appropriately broadened. Bandwidth broadening effectively prevents information loss.
The task of isolating octave components is not solely accomplished by the filter bank alone. Instead, its primary role is to provide a set of rich, frequency-focused input features. The eventual separation and classification are achieved by neural network models learning the combination patterns, spatial distributions, and relative importance of features across different frequency bands. The network effectively leverages these overlapping yet distinct sub-band information to learn discriminative feature representations for differentiating SSVEP targets. We briefly elaborate on the rationale for designing the filters in the “2.2 Data Preprocessing” section on page 3.
Reference [28] also bases its filter design on this theory.
Comments 2: Sensitivity, specificity, and F1 score for each of the four classes should be added in Table 1.
Response 2: We sincerely appreciate your valuable input on this matter. On two datasets, we analyze the sensitivity, specificity, and F1-score of all models across four categories under a 64-time window, as shown in Tables 2 and 3 on page 6. A detailed analysis is also presented in line 225 on page 7.

Reviewer 2 Report
Comments and Suggestions for Authors
The authors have addressed my concerns. I recommend the article for acceptance. Best of luck to the authors.
Author Response
We sincerely appreciate your thorough review and recognition of our work. Your valuable feedback has significantly improved the quality of our manuscript. Should you have further suggestions or require additional information, we are prepared to address them promptly.
We once again express our sincere appreciation for your support.

Reviewer 3 Report
Comments and Suggestions for Authors
Authors have addressed all reviewr's comments and the paper sounds much better now.
Author Response

(The authors gave the same response as above.)
